# KidSat: satellite imagery to map childhood poverty dataset and benchmark

## Abstract

Satellite imagery has emerged as an important tool to analyse demographic, health, and development indicators. While various deep learning models have been built for these tasks, each is specific to a particular problem, with few standard benchmarks available. We propose a new dataset pairing satellite imagery and high-quality survey data on child poverty to benchmark satellite feature representations. Our dataset consists of 33,608 images, each 10 km $\times$ 10 km, from 19 countries in Eastern and Southern Africa in the time period 1997-2022. As defined by UNICEF, multidimensional child poverty covers six dimensions and it can be calculated from the face-to-face Demographic and Health Surveys (DHS) Program [25]. As part of the benchmark, we test spatial as well as temporal generalization, by testing on unseen locations, and on data after the training years. Using our dataset we benchmark multiple models, from low-level satellite imagery models such as MOSAIKS [20], to deep learning foundation models, which include both generic vision models such as Self-Distillation with no Labels (DINOv2) models [15] and specific satellite imagery models such as SatMAE [6]. We provide open source code for building the satellite dataset, obtaining ground truth data from DHS and running various models assessed in our work.

## 1 Introduction

Major satellites like the Landsat or Sentinel program regularly circle the globe, providing updated, publicly available, high-resolution imagery every 1-2 weeks. An emerging literature in remote sensing and machine learning points to the promise that these large datasets, combined with deep learning methods, hold to enable applications in agriculture, health, development, and disaster response. A cross disciplinary set of publications hint at the potential impact, showing how satellite imagery can be used to estimate the causal impact of electrification on livelihoods [18], to measure income, overcrowding, and environmental deprivation in urban areas [23] and to predict human population in the absence of census data [27]. Despite these successes, machine learning for satellite imagery is not yet a well-developed field [19], with current approaches overlooking the unique features of satellite images such as variation in spatial resolution over logarithmic scales (from < 1 meter to > 1 km) [19] and the heterogeneous nature of satellite imagery in terms of the number of bands available from 3 bands for RGB to multispectral to hyperspectral.

Many areas of machine learning have advanced through the development of standardized datasets and benchmarks. Given the wide set of possible use cases for satellite imagery, there is no doubt room for multiple benchmarks. However there are only a few sources of up-to-date, high-quality satellite imagery, especially Landsat and Sentinel, so it is natural to construct publicly available datasets using these satellite programs.

Submitted to 38th Conference on Neural Information Processing Systems (NeurIPS 2024). Do not distribute.

Given the proven effectiveness of remote sensing for tasks that are naturally visible from space, such as land usage prediction, crop yield forecasting, and deforestation, we instead choose to focus on a more difficult task: multidimensional child poverty.

Of the 8 billion people in the world, over 2 billion are children (aged $< 18$ years old, as defined in the UN Convention on the Rights of the Child [24]). Child poverty is not the same as adult poverty; children are growing and developing so they have specific nutrition, health, and education needs—if these needs are not met, there can be lifelong negative consequences [3]. Poverty cannot simply be assessed by measuring overall household resources, as households may be very unequal and some of the needs of children, such as vaccines or education, may be neglected in households that are non-poor. Instead, child poverty must be measured at the level of the child and their experience [25].

Child poverty is based on the "constitutive rights of poverty" [25]. What this concept means is that child poverty includes important dimensions for children that require material resources to realize them, like education, health and nutrition, but exclude non-material dimensions such as neglect, violence, and lack of privacy. Crucially for the purposes of establishing a useful dataset and benchmark, the internationally agreed definition of child poverty was designed to enable cross-country comparisons [25].

While other benchmarks exist, most notably SUSTAIN-BENCH [29] which covers a range of sustainable development indicators, our newly proposed benchmark has the following features:

- We demonstrate the importance of fine-tuning transformer-based foundation vision models to tackle a challenging prediction task.
- Child poverty is both a multidimensional outcome, appropriate for fine-tuning large models, and a univariate measure (percent of children experiencing severe deprivation ranging from 0% and 100%) meaning that model performance can be intuitively grasped by policymakers.
- The amount of both satellite and survey data appropriate for child poverty prediction will continue to increase in the future, as UNICEF is now releasing geocodes as part of their Multiple Indicator Cluster Survey (MICS) program.

## 2 Related Work

### 2.1 Existing Satellite Imagery Datasets

With increased access to freely available high resolution satellite imagery through the Landsat and Sentinel programs, satellite image datasets have become very popular for training machine learning models. Models and datasets include functional map of the world (fMoW) [5], XView [14], Spacenet [26], and Floodnet [17] where the tasks are object detection, instance segmentation, and semantic segmentation. These are computer vision-specific tasks, rather than applied health and economic prediction problems, meaning the use of these datasets and models may be inappropriate for applied health and development researchers and practitioners.

### 2.2 Satellite Imagery for Demographic and Health Indicators

Machine learning models applied to satellite images are becoming more commonplace for analysing demographic, health, and development indicators as they can increase coverage by allowing for interpolation and faster analysis in under-surveyed regions. In an early work, satellite images were used to track human development at increasing spatial and temporal granularity [11]. Since then satellite images have been used to track development indicators which are clearly visible from space such as agriculture and deforestation patterns [2, 9, 28] but also more abstract quantities such as poverty levels [1], health indicators [7], and the Human Development Index [22].

### 2.3 Foundation Satellite Image Models

As increasing volumes of data become available, and with progress in self-supervised learning [10, 4], many foundation models are emerging. In computer vision, these large models are trained with self-supervised learning on hundreds of millions of images, serving as a "foundation" from which they can be fine-tuned for specific tasks. Popular examples of this are Vision transformers [8], CLIP [16], and DINO [15]. Recently, foundation models have been trained for satellite imagery specifically

on vast amounts of unlabelled satellite images. Examples of these are SatMAE [6] based on masked autoencoders, SatCLIP [13] based on CLIP [16], and DiffusionSat [12] which is a diffusion model [21] for generating satellite images. As it is not yet clear whether there is a benefit from training foundation models on more specific, but smaller datasets, we benchmark both generic foundation models for computer vision as well as satellite-specific foundation models.

# 3 Dataset

In this section, we introduce our unique dataset derived from the Demographic and Health Surveys (DHS) Program, combining high-resolution satellite imagery with detailed numerical survey data focused on demographic and health-related aspects in Eastern and Southern Africa. This dataset leverages the rigorous survey methodologies from DHS to offer high-quality data on health and demographic indicators, complemented by satellite images of the surveyed locations. The rich information embedded in the satellite images enables the application of advanced deep learning methods to estimate key poverty indicators in unsurveyed locations.

## 3.1 Satellite Images

This study utilizes high-resolution satellite imagery from two primary sources: Sentinel-2 and Landsat 5, 7, and 8. These satellite programs are chosen for their public accessibility, their specific advantages in computer vision applications, and their long history.

**Landsat 5, 7, and 8:** Part of a series managed by the United States Geological Survey (USGS), Landsat 5,7, and 8 together provide imagery with varying resolutions covering the time span from 1984 to the present (2024). Specifically, Landsat 8 captures data in 11 bands, including visible, near-infrared (NIR), and short-wave infrared (SWIR) at 30 meters resolution, panchromatic at 15 meters, and thermal infrared (TIRS) bands at 100 meters. This lower resolution for RGB bands contributes to space efficiency in data storage and processing, making it suitable for large-scale studies over extensive geographical areas. Additionally, the Landsat program, with missions dating back to 1972, provides an extensive historical archive of Earth imagery. This long timespan is particularly advantageous for our study as it allows the analysis of regions with survey data dating back to 1997.

**Sentinel-2:** Operated by the European Space Agency (ESA), Sentinel-2 features a multispectral imager with 13 spectral bands. The resolution varies by band: 10 meters for RGB and NIR, 20 meters for red edge and SWIR bands, and 60 meters for atmospheric bands. This high resolution in RGB bands provides richer information, which is valuable for large vision models requiring detailed visual data for accurate analysis. However, processing such high-resolution imagery can be computationally expensive, especially when dealing with large window sizes, posing challenges in terms of processing time and resource allocation. Additionally, Sentinel-2 only started collecting data in June 2015, which limits its use for analyzing events or changes that occurred before this date.

For each specified survey coordinate, we extract a 10 km × 10 km window of imagery using Google Earth Engine (GEE). Selection criteria for the imagery include the designation of a specific year and prioritization based on the least cloud cover within that year. This approach ensures that the images used are of the highest quality and most suitable for accurate analysis.

Both Sentinel-2 and Landsat series satellites include RGB bands, crucial for standard object recognition tasks in computer vision. Beyond the RGB spectrum, these satellites offer additional bands that are instrumental for advanced remote sensing analysis. This rich assortment of spectral data allows for sophisticated remote sensing techniques and predictive modeling, such as estimating vegetation density and water bodies, which are integral to our study on regional poverty estimation.

## 3.2 Demographic Health Surveys and Child Poverty

Dating back to 1984, the Demographic and Health Surveys (DHS) Program[1] has conducted over 400 surveys in 90 countries, funded by the US Agency for International Development (USAID) and undertaken in partnership with country governments. These nationally representative cross-sectional household surveys, with very high response rates, provide up-to-date information on a wide range of demographic, health and nutrition monitoring indicators. Sample sizes range between 5,000 and

---

[1]http://www.dhsprogram.com

30,000 households, and are collected using a stratified, two-stage cluster design, with randomly chosen enumeration areas (EAs) called "clusters" forming the sampling unit for the first stage. In each EA, a random sample of households is drawn from an updated list of households. DHS routinely collects geographic information in all surveyed countries. Cluster locations are released, with random noise added to preserve anonymity with this 'jitter' being different for rural and urban EAs.

The DHS data include both continuous and categorical variables, each requiring a different approach for aggregation to ensure accurate ecological analysis at the cluster level. For continuous variables, we calculated the mean of all responses associated with a particular spatial coordinate. Min-max scaling was applied after aggregation to normalise the data, ensuring that all values were on a scale from 0 to 1. Categorical variables were processed using one-hot encoding, which converts categories into binary indicator variables. Similarly, the mean of these binary representations was computed for each category at each cluster location.

Child poverty was assessed using a methodology formulated by UNICEF that evaluates child poverty across six dimensions: housing, water, sanitation, nutrition, health, and education. Each child was classified as moderately or severely deprived for each dimension based on a set of 17 variables in total [25]. An overall classification of moderate or severe deprivation is made if the child experiences moderate or severe deprivation on any of the six dimensions. Our target quantity of interest, `severe_deprivation`, was calculated as the percentage of children experiencing severe deprivation within a cluster. The detailed definition of moderate and severe deprivation and implementation of poverty calculation can be found in the supplementary material.

# 4 Benchmark

## 4.1 Spatial

We use 5-fold spatial crossvalidation at the cluster level across countries in Eastern and Southern Africa, spanning data collected from 1997 to 2022. We train our models on 80% of the clusters and evaluate its performance using the mean absolute error (MAE) of the `severe_deprivation` variable on the held-out 20% of clusters. This benchmark is designed to evaluate the model's capability to estimate poverty or deprivation levels at any given location based solely on satellite imagery data, quantifying the model's generalization capabilities to unsurveyed locations within surveyed countries.

## 4.2 Temporal

The temporal benchmark employs a historical data training approach, where we use data collected from 1997 to 2019 as the training set to develop our models. The objective is to predict poverty in 2020 to 2022. Model performance is evaluated using the MAE of the `severe_deprivation` variable. This benchmark tests the model's ability to capture temporal trends and forecast poverty based on satellite imagery data, assessing its forecasting accuracy. This capability is crucial for, e.g. nowcasting poverty before survey data becomes available.

## 4.3 Models to be Compared

We consider both baseline models (Gaussian process regression, mean prediction) and a range of more advanced computer vision models, both unsupervised and semi-supervised, with and without fine-tuning. Each model represents a distinct strategy in handling and processing satellite imagery:

**MOSAIKS** [20] is a generalisable feature extraction framework developed for environmental and socio-economic applications. We obtain MOSAIKS features from IDinsight, an open-source package that utilizes the Microsoft Planetary Computer API. The framework leverages satellite imagery to extract meaningful features from the Earth's surface. For our purposes, we used its Sentinel service, querying with specific coordinates, survey year, and a window size of 10 km $\times$ 10 km.

**DINOv2** [15] Initially designed for self-supervised learning from images, DINOv2 excels in generating effective vector representations from RGB bands alone. For our study, we selected the pre-trained base model with the vision transformer architecture as the backbone of our foundational model. We fine-tuned this foundational model with DHS variables to enhance its capability for predicting poverty. DINOv2 is evaluated in both its raw and fine-tuned forms using RGB imagery for both spatial and temporal benchmarks.

**SatMAE** [6] was originally developed for landmark recognition from satellite imagery. We fine-tuned with DHS variables to enhance its performance for predicting poverty. SatMAE has 3 variants: RGB, RGB+temporal, and multi-spectral. For benchmarking, we use the RGB variant for the spatial benchmark, and RGB+temporal for the temporal benchmark. The RGB+temporal variant takes 3 images of different timestamps from the same location; however, to facilitate a direct comparison with the other methods which use only a single image, we provide SatMAE with the same image three times. Additionally, it takes in Year, Month, and Hour, but since a DHS survey spans up to years, we only provide the Year variable, with Month and Hour set to January 00:00.

### 4.3.1   Evaluation and Fine-tuning

In our fine-tuning pipeline, we start from DINOv2's and SatMAE's original checkpoints with an uninitialised head and train it against 17 selected DHS variables to minimize mean absolute error (MAE). We then evaluate the model by replacing the head with a cross-validated ridge regression model mapping satellite features to the `severe_deprivation` variable and calculate the MAE loss on a test set that was neither seen by the fine-tuned model nor the Ridge Regression. For the spatial task, we perform a 5-fold cross-validation on the whole dataset, and for the temporal benchmark, we take the training set as the data before the year 2020 and evaluate on the data from 2020 to 2022.

For the spatial benchmark, we randomly split the data into five train-test splits using a reproducible script. For the temporal benchmark, we divided the data into a single fold, using data from before 2020 for training and data from 2020 onward for testing.

For DINOv2, we used a batch size of 8 for Landsat imagery and a batch size of 1 for Sentinel imagery, with L1 loss and an Adam optimiser of learning rate and weight decay both set to 1e-6. We trained the model for 20 epochs with Landsat imagery and 10 epochs with Sentinel imagery, selecting the model with the minimum validation loss on predicting the 17 DHS variables. Each task was trained on a single Nvidia V100 32GB GPU, with an average training time of 1 hour per epoch for Landsat and 2 hours per epoch for Sentinel imagery. For SatMAE, we resize the input to $224 \times 224$ and use a batch size of 64 for the spatial task and 32 for the temporal task. Training is done with Adam optimiser with learning rate 1e-5 and weight decay 1e-6, for at most 20 epochs with the early stopping of patience 5 and delta 5e-4. Each task is trained on a single Nvidia L4 GPU, taking, for Landsat and Sentinel, 1 and 2 hours for the first epoch and 15 and 10 minutes for each subsequent ones with data caching.

## 5   Results

The performance of the child poverty prediction models is summarized in Table 1.

### 5.1   Spatial Benchmark

In the spatial benchmarking, Gaussian Process regression with geographic coordinates resulted in a mean absolute error (MAE) that is 0.04 lower than that achieved by the baseline mean prediction model. Notably, regressions using outputs from foundational vision models outperformed both the baseline and GP regression. The MOSAIKS features based on Sentinel-2 imagery achieved 0.2356 MAE on predicting the `severe_deprivation` variable. Utilising Landsat imagery, the DINOv2 and SatMAE achieved MAEs of 0.2260 and 0.2341 respectively. Further enhancements through fine-tuning with Demographic and Health Surveys (DHS) variables led to reduced prediction errors, with DINOv2 and SatMAE recording MAEs of 0.2042 and 0.2125 respectively. When using Sentinel-2 imagery, the SatMAE architecture achieved errors of 0.2347 and 0.2093 before and after the fine-tuning, while DINOv2 further lowered the errors to 0.2013 and 0.1836 respectively.

### 5.2   Temporal Benchmark

In the temporal benchmark, models faced greater challenges in forecasting poverty. Gaussian Process regression was substantially worse than the mean prediction. Using Sentinel-2 imagery, MOSAIKS recorded an MAE of 0.2588, with DINOv2 and SatMAE achieving MAEs of 0.2597 and 0.3067 respectively. Additional fine-tuning with DHS variables led to increased prediction errors, with DINOv2 and SatMAE resulting in MAEs of 0.2858 and 0.3139. Employing Landsat imagery, the pre-trained DINO v2 and SatMAE model achieved worse initial MAEs of 0.2704 and 0.3453;

Table 1: Comparison of MAE on `severe_deprivation` across Benchmarks and Imagery Sources. In the spatial task, random clusters are heldout, while the temporal task is a more difficult forecasting task, with the years 2020-2022 held out. Fine-tuning consistently gives better results. While SatMAE is a foundation model trained on satellite imagery, it is outperformed by the more generic DINOv2 foundation model.

| Model | Benchmark Type | MAE ± SE (Spatial) | MAE (Temporal) |
|---|---|---|---|
| Mean Prediction | - | 0.2930 ± 0.0018 | 0.3183 |
| Gaussian Process Regression | - | 0.2436 ± 0.0002 | 0.5656 |
| MOSAIKS | Sentinel-2 | 0.2356 ± 0.0114 | 0.2588 |
| DINOv2 (Raw) | LandSat | 0.2260 ± 0.0005 | 0.2704 |
| DINOv2 (Raw) | Sentinel-2 | 0.2013 ± 0.0019 | 0.2597 |
| DINOv2 (Fine-tuned) | LandSat | 0.2042 ± 0.0015 | 0.2574 |
| DINOv2 (Fine-tuned) | Sentinel-2 | 0.1836 ± 0.0036 | 0.2858 |
| SatMAE (Raw) | LandSat | 0.2341 ± 0.0017 | 0.3453 |
| SatMAE (Raw) | Sentinel-2 | 0.2347 ± 0.0027 | 0.3067 |
| SatMAE (Fine-tuned) | LandSat | 0.2125 ± 0.0019 | 0.3376 |
| SatMAE (Fine-tuned) | Sentinel-2 | 0.2093 ± 0.0039 | 0.3139 |

nevertheless, additional fine-tuning on DHS variables resulted in relative equal performance for both models, with MAEs of 0.2574 and 0.3376 respectively.

## 5.3 Interpretation of Results

The performance of various poverty prediction models is shown in Table 1. Our prediction task is the percentage of a location's children who are experiencing severe deprivation, so a MAE on the order of 0.20 is equivalent to 20 percentage points of error, which policymakers may consider to be too high to be useful. The spatial benchmark demonstrates the advantage of using foundational vision models over the baseline mean prediction model and GPR. Models like MOSAIKS, DINOv2, and SatMAE, particularly when improved through fine-tuning with DHS variables, show a further reduction in mean absolute error (MAE). This implies that spatial features extracted from satellite imagery are comparably more effective than GP modelling in estimating poverty indicators in regions where surveys have not been conducted.

The temporal benchmark, which evaluated a forecasting task (predict 2020-2022 using data from before 2020), was more difficult than the spatial benchmark. Satellite imagery is at best a proxy for multidimensional child poverty, and this finding suggests it is a better proxy for quantifying spatial as opposed to temporal variation. Satellite imagery models performed worse on the temporal as compared to spatial benchmark, and the fine-tuned models, particularly those using Sentinel-2 imagery as the source input, showed increased MAE compared to the raw output from both DINOv2 and SatMAE models. This suggests that the models overfit the historical data, and struggled to generalise to data collected after 2020. Gaussian process regression based on spatial coordinates had no way of predicting changes over time, explaining its very poor performance.

## 6 Discussion and Future Work

### 6.1 Satellite Imagery Sources

As compared to Landsat, models utilising Sentinel-2 imagery, such as the fine-tuned versions of DINOv2 and MOSAIKS, demonstrated improved performance in both spatial benchmarks. These models benefited from the rich spectral information provided by Sentinel-2, which enabled more precise predictions of deprivation levels across diverse geographical regions.

Additionally, the computational demands associated with processing high-resolution Sentinel-2 data present substantial challenges. For instance, large versions of vision transformers could not be accommodated within the memory constraints of a 32 GB GPU when processing the full Sentinel-2 data. In contrast, these larger models could be deployed with Landsat data, which offers lower resolution but requires less computational resources. Under the spatial setting, this scenario highlights

a critical trade-off in model deployment: the choice between employing lightweight models to retain the high resolution of Sentinel-2 imagery or opting for more powerful models that necessitate a reduction in image resolution to ensure feasibility.

## 6.2 Modeling Choices

We considered a representative set of models: MOSAIKS is unsupervised, DINOv2 is a generic foundation model trained on images, and SatMAE is a foundation model trained on satellite imagery.

### 6.2.1 MOSAIKS

MOSAIKS is designed to provide general-purpose satellite encodings and is notably accessible through Microsoft's Planetary Computer service. This model generates a large output vector, typically around 4000 dimensions, which, while comprehensive, can lead to significant computational costs when methods beyond simple linear regression are employed. Furthermore, although MOSAIKS is well-suited for broad applications, integrating online feature acquisition into a fine-tuning process tailored specifically to poverty prediction presents challenges. This limitation can hinder its effectiveness when adapting to specific tasks where dynamic feature updates are crucial. We also note that MOSIAKS' API at times returned no-features, even after implementing rate-limiting mechanisms. This random behaviour combined with unavailability of features before 2013 limits the use of MOSAIKS considerably.

### 6.2.2 DINOv2

DINOv2 stands out as a state-of-the-art foundational model that excels in generating effective vector representations from RGB bands alone, achieving comparable performance to models that utilize additional spectral bands. Its flexibility in model sizing allows users to select the optimal model scale for specific training needs, enhancing its utility across various computational settings. The availability of pre-trained weights simplifies the process of fine-tuning for specialized tasks such as poverty prediction. However, DINOv2's reliance solely on RGB bands means it does not leverage the broader spectral information available in other satellite imagery bands, which may limit its application scope to scenarios where such data could provide additional predictive insights.

### 6.2.3 SatMAE

SatMAE demonstrates respectable results, surpassing baseline models even with only its raw, pre-trained configuration. Its architecture inherently supports the integration of multispectral and temporal analysis, making it well-suited for handling complex datasets typically encountered in satellite imagery analysis. Despite these strengths, the pre-trained SatMAE model is configured to process images of $224 \times 224$ pixels, constraining its ability to utilize higher-resolution imagery, such as the $1000 \times 1000$ pixel images from Sentinel-2. This limitation restricts its performance, particularly in comparison to models that can fully exploit high-resolution data, thereby failing to match the effectiveness of other advanced models in our analysis. Another limitation is that with our simple benchmarking setup, we have not made full use of SatMAE's temporal and multi-spectral capabilities. In the temporal setup, we are providing only one image-timestamp pair with only the Year variable, while the model is capable of taking up to 3 pairs, along with Month and Hour variables. We are also exclusively using the RGB bands, while the multi-spectral version of SatMAE is capable of taking in other bands of the satellite images in our dataset.

## 6.3 Further Discussion

The ability to accurately measure poverty across a vast number of geolocations is crucial for understanding and addressing the disparities that exist in different regions. The extensive and high-quality poverty measurement is valuable for researchers and policymakers. It allows for the analysis of poverty trends and the effectiveness of current policies, thereby facilitating more informed decision-making to reduce global poverty.

Traditional surveys, while rich in data, are limited by geographical and logistical constraints. Conducting extensive on-the-ground surveys is not only costly but also time-consuming—from data collection to processing and harmonisation. In regions lacking detailed survey data, traditional methods like

GPR or nearest-neighbor approaches are typically used to estimate poverty levels. However, these methods can be unreliable, particularly when extrapolating data to locations far from surveyed areas or data with temporal dependencies, leading to high uncertainty.

On the other hand, satellite imagery, which was made widely available by organizations such as the European Space Agency and the United States Geological Survey, can be accessed from any geographic location. Recent advancements in the field of computer vision have made it possible to infer meaningful information from this imagery, which can effectively improve poverty prediction. By demonstrating the capabilities of large vision models and satellite imagery in this context, we aim to inspire and encourage others in the field to further develop and refine these methods, thus driving changes in sociology research and policy making.

### 6.4 Limitation and Future Directions

Our study had a number of limitations. While high-quality household survey data is expensive to acquire, it is an irreplaceable source of ground truth; machine learning can complement and enhance, but never replace, these datasets. We highlighted the difficulty of the temporal benchmark, suggesting that future research could explore time series methods for forecasting or ways of better encoding temporal information into the foundation models. Another limitation of our study is that, in our goal of learning general representations of satellite imagery, we fine-tuned the large vision models to predict DHS variables important to calculating the `severe_deprivation` variable rather than directly optimising the models on the `severe_deprivation` variable itself. While this approach could lead to better generalisation by leveraging a broader range of demographic and health indicators, it may not be the most effective strategy for minimising the MAE specifically for `severe_deprivation`. This indirect optimisation can result in sub-optimal performance for the target variable. Future work could explore the extent of benchmark improvement with direct optimisation techniques for `severe_deprivation`. Additionally, future work could also evaluate the performance of various models on the six components of child poverty separately, on moderate as opposed to severe deprivation. While we evaluated spatial generalisation by leaving out clusters, a stricter evaluation would have considered a leave-one-country out evaluation.

## 7 Conclusion

In conclusion, our study demonstrates the potential of satellite imagery combined with large vision models to estimate child poverty across spatial and temporal settings. We introduced a new dataset pairing publicly accessible satellite images with detailed survey and child poverty data based on the Demographic and Health Surveys Program, covering 19 countries in Eastern and Southern Africa over the period 1997-2022. By benchmarking multiple models, including foundational vision models like MOSAIKS, DINOv2, and SatMAE, we assessed their performance in predicting child poverty. Our results show that advanced models with satellite imagery have the potential to outperform baseline methods, offering more accurate and generalisable poverty estimates. This work highlights the importance of integrating remote sensing data with machine learning techniques to address complex socio-economic issues, providing a scalable and cost-effective approach for poverty estimation and policy-making.

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
