# A    Appendix / supplemental material

## A.1    Dataset description

The KidSat dataset we present in this work includes both cluster-wise child poverty derived from the DHS data and the satellite imagery corresponding to each cluster. Due to the confidentiality of the survey data, DHS requires registration prior to accessing the data. We include detailed procedures for acquiring the satellite imagery and DHS data in our GitHub repository. For the review process, we provide the data and pre-trained models: `https://drive.google.com/drive/folders/1W9-bFdf5FEGG8J3U6Z9KcZ8I2iSG4lGm?usp=sharing`. The access will be removed after the review.

### A.1.1    Coding child poverty

The `severe_deprivation` variable is used in this work to represent severe child poverty for individual responses within the cluster. It is calculated by aggregating several indicators of severe deprivation across multiple dimensions such as housing, water, sanitation, nutrition, health, and education. A child is considered severely deprived if they experience at least one severe deprivation in any of the following areas:

1. Housing: Severe deprivation if the number of persons per room is 5 or more.
2. Water: Severe deprivation if the household uses unsafe water sources.
3. Sanitation: Severe deprivation if the household lacks access to safe sanitation facilities.
4. Nutrition: Severe deprivation if a child's height-for-age z-score is below -3 (indicating severe stunting).
5. Health: Severe deprivation if a child misses all essential vaccines or has untreated acute respiratory infections.
6. Education: Severe deprivation if a school-aged child does not attend school and has not completed any level of education.

Among all DHS variables involved in child poverty calculation, we selected 17 variables, presented in Table 1, as the prediction objective during model fine-tuning. We scaled the continuous variables to the range from 0 to 1. We expanded the categorical columns using one-hot encoding, where each column is in the format of `variable_value`. In total, a vector of dimension 99 based on these 17 DHS variables was used to represent a cluster, where we mapped the satellite imagery to predict these vectors as the method of model fine-tuning.

## A.2    Compute

As one of the heavy-lifting parts is loading images, a multi-core CPU ($\geq 8$) is recommended to optimise the data loading using multiple workers with the data loader. The training was done using Nvidia Tesla V100 GPUs for DINO v2 experiments and Nvidia L4 GPUs for SatMAE experiments. In particular, for DINO v2 experiments with Sentinel imagery, 32 GB of GPU memory is a hard requirement.

## A.3    Experimental details

The dataset for our experiments was sourced from Sentinel-2 and Landsat-8 satellite imagery. Each image tile, corresponding to specific geographical centroids, was preprocessed by selecting and stacking RGB bands, normalising, and clipping pixel values to the 0-255 range. Data was organised into cross-validation folds with separate training and testing sets. Only centroids with available imagery were included, and rows with missing target values were excluded to maintain data integrity. As a MOSAIK

### A.3.1    DINOv2

We used the base DINOv2 Vision Transformer (ViT) model which has an output of 768-dimension feature vector. We appended a regression head for mapping the feature vector to the 99-dimension

Table 1: DHS variables selected for model fine-tuning

| Variable | Description | Type |
|---|---|---|
| h10 | Whether the child ever received any vaccination to prevent diseases. | Categorical |
| h3 | DPT 1 vaccination. | Categorical |
| h31 | Whether the child had suffered from a cough in the last two weeks and whether the child had been ill with the cough in the last 24 hours. | Categorical |
| h5 | DPT 2 vaccination. | Categorical |
| h7 | DPT 3 vaccination. | Categorical |
| h9 | Measles 1 vaccination. | Categorical |
| hc70 | Height for age standard deviation (according to WHO). | Continuous |
| hv106 | Highest level of education the household member attended. | Categorical |
| hv109 | Educational attainment recoded. | Categorical |
| hv121 | Household member attended school during current school year. | Categorical |
| hv201 | Main source of drinking water for members of the household. | Categorical |
| hv204 | Time taken to get to the water source for drinking water | Continuous |
| hv205 | Type of toilet facility in the household. | Categorical |
| hv216 | Number of rooms used for sleeping in the household. | Continuous |
| hv225 | Whether the household shares a toilet with other households. | Categorical |
| hv271 | Wealth index factor score (5 decimals) | Continuous |
| v312 | Current contraceptive method. | Categorical |

47 target DHS variables. The Adam optimiser was used, with a learning rate and weight decay of 1e-6
48 for both the base model and the regression head. Training involved iterating over 20 epochs for
49 Landsat imagery and 10 epochs for Sentinel imagery to minimise the L1 loss between predicted and
50 actual target values. The batch size was determined by available GPU memory, 8 for Landsat imagery
51 source and 1 for Sentinel. The average training time is 1 hour per epoch for Landsat experiments and
52 2 hours per epoch for Sentinel experiments.

### A.3.2 SatMAE

54 The base SatMAE model is the decoder of a MAE, outputting a 1024-dimensional vector. Fine-tuning
55 is done by appending a transformer head mapping to the 99-dimensional target DHS variables. We
56 use the Adam optimiser, with a learning rate of 1e-5 and weight decay of 1e-6 for both the base model
57 and the head. Training involved iterating over 20 epochs for both Landsat imagery and Sentinel
58 imagery to minimise the L1 loss between predicted and actual target values, with early stopping of
59 patience 5 and delta 5e-4. The batch size is 32 for spatial task and 16 for temporal task. Training
60 takes 1 hour for the first epoch and 15-30 minutes for each subsequent one when data caching is used.