# OpenReview forum: "KidSat: satellite imagery to map childhood poverty dataset and benchmark"
_NeurIPS.cc/2024/Datasets_and_Benchmarks_Track — Submitted to NeurIPS 2024 Track Datasets and Benchmarks_

### Official Review · Reviewer_rwhX · 2024-07-18
**Interesting task and dataset, flawed baselines**

**Rating:** 5
**Confidence:** 4
**Clarity:** Quite clear and well written.

**Review:**

**Strength:**
- The task is interesting, impactful, and challenging, as well as relatively new in the ML community.
- The authors consider several baselines and selected two relevant foundation models (one from vision and one from Earth observation).
- The paper is generally well-written and effectively presents and motivates the task and data.

**Weaknesses:**
1. Some dubious choices have been made in the design of baselines and experiments, invalidating the conclusions. For example, the paper considers images that are (i) high-resolution, (ii) multi-spectral, and (iii) multi-temporal, but (i) severely downsamples them, (ii) only uses RGB bands, and (iii) ignores the temporal aspect by repeating the same image several times.
2. Some claims are vague and/or not validated experimentally.
3. Figures and graphs would strengthen the paper (e.g., spatial distribution of data, results). This is particularly concerning as the authors have over one entire unused page.
4. The target variable is likely very spatially correlated, and the models are probably mostly performing (indirect) geolocation. Some experiments and metrics should address this point. The spatial autocorrelation is also not addressed in the train/test split.

Details about these weaknesses are given in the Improvement section.

**Overall**: This paper introduces an interesting dataset for a valuable and challenging task. However, several design flaws in the dataset and the implementation of the methods limit the value of the analysis. Moreover, the total lack of visuals (which are sorely missing to convey crucial information) and subpar tables, combined with the remaining space and obvious missing experiments (several of which are alluded to in the conclusion), convey the impression that the paper has been rushed and unfinished.

The dataset has a lot of value, and once fixed, the experiment will too. The subject is important and deserves a finished paper. The reviewer encourages the authors to resubmit the paper once it is ready.

**Strengths:**

- The task is interesting, impactful, and challenging, as well as relatively new in the ML community.
- The authors consider several baselines and selected two relevant foundation models (one from vision and one from Earth observation).
- The paper is generally well-written and effectively presents and motivates the task and data.

**Additional Feedback:**

N/A

**Correctness:**

The paper is correct in the sense that nothing is false, but the baselines (and SatMAE in particular) are misused, and some claims are not validated.

**Documentation:**

Good

**Ethics:**

The authors took some steps to preserve privacy. I don't have concerns about ethics. However, since the dataset directly refers to sensitive information about vulnerable people (poor children), filling out a DataSheet for DataSet or something similar would have been helpful.

**Limitations:**

Addressed

**Opportunities For Improvement:**

1. **Dubious design choice**
> We use the RGB variant [of SatMAE].
- Why? The ability of SatMAE to handle multispectral images is a strong advantage, which should be quantified. The authors collected multispectral data but never used any spectral bands beyond RGB.
> The RGB+temporal variant takes 3 images of different timestamps from the same location; however, to facilitate a direct comparison with the other methods which use only a single image, we provide SatMAE with the same image three times.
- Same concern. The ability of SatMAE to handle multitemporal data is a strong advantage, which should be quantified.
>  Additionally, it takes in Year, Month, and Hour, but since a DHS survey spans up to years, we only provide the Year variable, with Month and Hour set to January 00:00
- Why? You have the precise time of the satellite acquisition.
> Future work could [directly optimizing for the target variable] severe_deprivation
- That seems like a very simple experiment, which addresses an unnecessary weakness of the benchmark: the fine-tuning task is different from the evaluation task.
> future work could also evaluate the performance of various models on the six components of child poverty separately
- That would also be a simple experiment and give more depth to the analysis. Adding moderate poverty as a target would also be valuable.
> Gaussian process regression
- No detail at all, what is the input of this method?
> we used a batch size of 8 for Landsat imagery and a batch size of 1 for Sentinel imagery
- The images are not resized for DINO? Are you feeding 1000x1000 images to the loader? If yes, that is a waste of resources as DINO will resize them to 224x224 anyway. You could use the same batch size as SatMAE.

2. Some claims are vague and/or not validated experimentally
> We demonstrate the importance of fine-tuning transformer-based foundation vision models
- The only non-baseline models evaluated are all "fine-tuned" & "transformer-based" & "foundation models." To validate this claim, the authors should consider an alternative to fine-tuning (with LoRA, for example), to transformers (e.g., ResNet50), and not "foundation models," such as a ViT pretrained "just" on ImageNet.
> additional bands are instrumental for advanced remote sensing analysis
- this is not evaluated

3. Figures and graphs
- A map with the location of the clusters would be very informative. It could, for example, represent the train/test split (and answer some questions below) and show the distribution of "severely/moderately" deprived areas and the "culprit" variables (housing, water, sanitation, nutrition, health, and education, or a combination).
- Showing examples of images from Sentinel-2 and Landsat would help appreciate the difficulty of the task and the benefit of the resolution.
- Table 1 is very hard to read as (i) there are too many digits, and (ii) there are too many dimension changes (backbone, fine-tuning, sensor). A well-designed bar plot would make the main paper much clearer.

4. Spatial correlation.
- The data may be highly spatially correlated, and the task may boil down to a "simple" geolocation task. To measure this, the authors should measure the correlation between location and the target variables and plot semivariograms. They could also consider a baseline that only uses the location as input. The proposed experiment in the conclusion of having a left-out country is also valid.
-  The spatial autocorrelation is also not addressed in the train/test split. Are there close clusters in both splits?

**Relation To Prior Work:**

Ok

**Summary And Contributions:**

This paper introduces a benchmark for regressing a child poverty index from satellite images. The authors evaluate several baselines and models on this task, demonstrating its difficulty.

---

> ### Author Rebuttal · Authors · 2024-08-16
>
> We thank the reviewer for their constructive comments. Overall, we agree that there is space for improved figures and tables, and further evaluations. We commit to making changes as outlined below. (Please see the responses to the other reviewers about a new figure and a new table.) We will also follow your suggestion of including a Datasheet for our dataset.
>
> As discussed in our response to the first reviewer, we believe the strength of our paper is in proposing child poverty prediction as an important new benchmark and dataset, inviting deep learning experts to solve this task, rather than providing the most state-of-the-art method ourselves. In particular, fully exploring SatMAE’s specific suitability to our task is a worthy goal, but it is beyond the scope of our paper.
>
> We respond point-by-point to your opportunities for improvement below.
>
> *1. Dubious design choice*
>
> For consistency across computer vision models and satellite imagery we provided all models (DINO, SatMAE, and MOSAIKS) with RGB imagery only. This setup could put SatMAE at a disadvantage, especially before fine-tuning, since it was pretrained with Sentinel imagery with 10 spectral bands. We will add this as a limitation to our discussion.
>
> Regarding the temporal data, you are correct that satellite imagery has exact timestamps, and SatMAE might perform better when provided with three satellite images. However, the survey data is not precisely timestamped as it was collected across multiple months. And again, including extra images for SatMAE complicates the simplicity and generality of our dataset and benchmark.
>
> Of course, future work could use the baselines we have established, and extra imagery, to evaluate SatMAE on its own terms, and hopefully lead to improved versions. To this end, as discussed in the response to reviewer 1, we have now included a well-documented workflow to show how to evaluate other methods.
>
> Regarding the use of severe_deprivation directly as the target variable and also the performance on the six components of child poverty, we absolutely agree, and will add these evaluations to the final version of the paper.
>
> Regarding the Gaussian process regression setup, please see our below discussion of spatial correlation.
>
> Regarding image resizing, we are not using the complete DINOv2 code, but using the trained ViT backbone from DINO. Before processing, the image is not resized and the full 1000 x 1000 image is passed through the model allowing us to leverage the high resolution available in our imagery.
>
> *2. Some claims are vague and/or not validated experimentally*
>
>
> First, we note that we have included MOSAIKS, a random feature representation for satellite imagery which other literature has demonstrated to be very effective. MOSAIKS is not a foundation model, nor is it fine-tuned or transformed based. We agree that alternatives to fine-tuning such as LoRA would be very interesting to consider, but leave that for future work. Regarding alternatives to transformers, we have now run a comparison with ResNet50 (spatial benchmark performance 0.2399 ± 0.0015) which we will include in the table.
>
> *3. Figures and graphs*
>
> We agree with your suggestions and will follow them as best as we can:
>
>
> Please see the new Figure 1 attached.
>
> DHS cluster locations, even with jitter added, are considered sensitive data, so they cannot be visualized directly (https://dhsprogram.com/pubs/pdf/SAR7/SAR7.pdf), but visualizing the train/test split is a very good idea. As shown in the attached figure, which we will include in the supplementary materials, train/test cluster locations are often very close in space.
>
> Your idea of showing images from Sentinel-2, Landsat, and visualizing the subindicators (“culprit variables”) are all very good ones, which we will follow. We will include Sentinel-2 and Landsat images in the main text if space allows, and subindicators in the supplementary materials.
>
> We will follow your suggestion of reformatting the table for clarity and adding a barplot in the main text.
>
> *4. Spatial correlation.*
>
> As shown in the attached figures, spatial correlation is indeed a concern: poverty varies smoothly in space (Figure 1) and train and test cluster locations can sometimes be very close (Supplementary Figure 1). On the other hand, we included Gaussian process regression (also known as kriging) as a baseline method. We have added text explaining that Gaussian process regression was performed with latitude/longitude coordinates as inputs and severe_deprivation as the target variable. This approach is a standard spatial statistics approach to georeferenced data of this type, and by their construction, Gaussian processes specifically leverage spatial correlation in order to provide accurate interpolation at out-of-sample locations. But as shown in Table 1, all satellite methods outperform Gaussian process regression, from which we conclude that spatial correlation alone does not suffice to provide accurate interpolation.
>
> To further explore the issue of spatial correlation, and mimic the situation in which we need to make predictions for an entirely unsampled country, we will add a leave-one-country out analysis to the paper. Generalization to an unseen country is a very strict evaluation of our method, not subject to spatial autocorrelation.
>
> To conclude, we are very happy to follow the suggestions of the reviewer in terms of improving the clarity of the paper and strengthening the evaluations.

---

> > ### Comment · Reviewer_rwhX · 2024-08-20
> > **Thanks for clarification**
> >
> > The reviewer thanks the authors for their detailed response and appreciates the following improvements:
> > - The inclusion of additional models in the evaluation.
> > - Clarifications on the Kriging approach, which is a solid baseline.
> > - The commitment to adding the promised graphs, analyses, and experiments.
> >
> > However, the following concerns remains:
> > - The reviewer respectfully disagrees with the authors' assertion that providing a remote sensing model with comprehensive geospatial information (including all spectral bands, a temporal stack, and appropriate pretraining) is unfair. On the contrary, utilizing this information would not complicate the pipeline but rather guide future benchmark users on how to fully leverage the available data.
> > - Additionally, it’s worth noting that some models are not directly comparable as they currently stand. For example, DINO processes 1000x1000 images, while SatMAE processes 224x224 images. This discrepancy should be considered in the evaluation.
> > - The reviewer is also concerned about the high degree of spatial correlation in the current data splits. Implementing a clean split with a buffer zone would mitigate this issue and lead to more robust evaluation results.
> >
> > In conclusion, while the reviewer is inclined to raise the review to a borderline reject, given the originality and importance of the subject, there are still limitations in the benchmark's design and evaluation, and the amount of promised work is significant. The reviewer believes that the paper would greatly benefit from cleaner evaluations and improved presentations, alongside all the promised analyses and experiments.
> >
> > The reviewer would not strongly object to the paper's acceptance but feels that the subject matter deserves a more polished presentation and rigorous evaluation.

---

> > > ### Author Response · Authors · 2024-08-27
> > >
> > > We would like to thank the reviewer for their quick response and consideration of our rebuttal.
> > > We would like to address some of the concerns that remain :
> > >
> > > *1. Geospatial Data*
> > >
> > >
> > > While we acknowledge the potential significance of incorporating full geospatial data to enhance the benchmark's performance, it would necessitate an entirely new set of experiments. This falls outside the scope of our current project, which primarily focuses on creating a dataset, establishing a benchmark for an important task, and providing baselines.
> > >
> > > Nevertheless, we note to incorporate the full geospatial data as a future work that we are committed towards.
> > >
> > > *2. Image Size*
> > >
> > > To address issues around not providing the same size of images to SatMAE and Dinov2, we ran another experiment where images were resized to 224x224 before being passed into Dinov2. The result MAEs were as follows :
> > >
> > > |                                          | Spatial            | Temporal |
> > > |------------------------------------------|--------------------|----------|
> > > | Resize input 224 deprived_sev (Landsat) | 0.2169±0.0017      | 0.2781   |
> > > | Resize input 224 deprived_sev (Sentinel)  | 0.2018±0.0028      | 0.3030   |
> > >
> > > We note that performance is worse than before (1000 x 1000). However, Dinov2  still slightly outperforms SatMAE. We acknowledge that the current SatMAE cannot process larger images because it was trained on 224x224 images.
> > >
> > > Hence, passing larger images to DINO would create an unfair comparison. Nevertheless, it also shows us that using high-resolution images increases the performance. We will include this information in the results section and make a note of it in the paper for clarity.
> > >
> > > *3. Spatial correlation*
> > >
> > >
> > > While we acknowledge that spatial correlation presents a challenge, the improved performance from the  kriging baseline (a standard spatial statistics model) suggests that the satellite imagery based models are acquiring additional information.
> > >
> > >
> > > To test spatial correlation more strictly, we have added the country leave one out split. We show performance averaged over 16 splits, where each split leaves out one of the countries to test, and trains on the remaining 15. The performance is as follows :
> > > |                                           | One-vs-rest split  |
> > > |-------------------------------------------|--------------------|
> > > | DINOv2(Landsat)         | 0.2619±0.0124      |
> > > | DINOv2(Sentinel)          | 0.2428±0.0147      |
> > >
> > > **. Additional experiments**
> > >
> > >
> > > We have also performed other experiments :
> > > Using DINOv2 with Sentinel imagery we have tried to learn all the sub-indicators of poverty at both the moderate and the severe level
> > > |                            | Spatial          | Temporal |
> > > |----------------------------|------------------|----------|
> > > | dep_education_moderate      | 0.0133±0.0001    | 0.0125   |
> > > | dep_education_severe        | 0.0128±0.0002    | 0.0120   |
> > > | dep_health_moderate         | 0.0730±0.0013    | 0.1184   |
> > > | dep_health_severe           | 0.0371±0.0005    | 0.0559   |
> > > | dep_housing_moderate        | 0.1620±0.0022    | 0.2184   |
> > > | dep_housing_severe          | 0.1239±0.0015    | 0.1827   |
> > > | dep_nutrition_moderate      | 0.1082±0.0007    | 0.1325   |
> > > | dep_nutrition_severe        | 0.0579±0.0004    | 0.0713   |
> > > | dep_sanitation_moderate     | 0.1918±0.0013    | 0.2636   |
> > > | dep_sanitation_severe       | 0.1941±0.0024    | 0.3031   |
> > > | dep_water_moderate          | 0.2221±0.0011    | 0.2648   |
> > > | dep_water_severe            | 0.2126±0.0026    | 0.2461   |
> > >
> > > This is an interesting experiment which shows which sub-indicators are easier to learn and which are harder.
> > >
> > > As requested, we also tried to directly optimise for the objective both at the severe and the moderate level.
> > >
> > > | | Spatial | Temporal |
> > > |---|---|---|
> > > | Directly optimizing deprived_sev (Sentinel) | 0.1872+-0.0021 | 0.2688 |
> > > | Directly optimizing deprived_mod (Sentinel) | 0.1082+-0.0011 | 0.1401 |
> > > | Directly optimizing deprived_sev (Landsat) | 0.2114+-0.0011 | 0.2779 |
> > > | Directly optimizing deprived_mod (Landsat) | 0.1103+-0.0010 | 0.1418 |
> > >
> > > We can see the performance remains similar to ones reported in the paper, which shows that direct optimization is not the key to improving performance.
> > >
> > > In the revised manuscript, we aim to incorporate all the newly conducted experiments and their outcomes. We anticipate that these experiments will address the concerns raised by the reviewer in the earlier review.
> > >  We express our gratitude to the reviewer for their valuable insights and assistance in enhancing the quality of our paper.

---

### Official Review · Reviewer_yepH · 2024-07-23

**Rating:** 7
**Confidence:** 4
**Correctness:** The claims and dataset construction a…

**Review:**

This is an important problem. The paper is clear, it is well written. The presented dataset would be an important dataset in the important problem. The dataset curation and licenses all look appropriate. The appendix fills in some of the important details about the domain missing from the main paper. The benchmark tasks are reasonable and are a valuable addition (though they take space and thus the dataset part gets squeezed a bit!). The discussion and the conclusions from the benchmarks are important and point to the future scope on this dataset and problem.

**Strengths:**

- This paper tackles an important problem and importantly releases a large dataset
- The contribution of the baselines is valuable to set the benchmark. The presented benchmarks appear reasonable.
- The paper is well-written and is easy to follow along
- The code/dataset provided seems reasonably easy to follow along and thus might make it easier to enter in this domain.
- The comment and discussion on what satellite product to use is an insightful discussion.

**Additional Feedback:**

NA

**Clarity:**

The paper sectioning is easy to follow along. The written text flows well, is easy to digest.

**Documentation:**

The repository included does seem to provide enough information to replicate this dataset and benchmarks

**Ethics:**

I believe that the data is at an aggregated level and thus not subject to ethical concerns.

**Limitations:**

I think some interpretability and explainability analysis and then showing the results to domain experts would be useful. Else, I believe some places may be hard done by as they may be missed by the presented ML models.

**Opportunities For Improvement:**

1. Perhaps the dataset size and diversity can be explained in more detail -- this will help emphasise the contribution over prior art which seems to be smaller and less diverse.
2. Perhaps some more discussion on the dataset attributes (which have been nicely explained in the appendix) can be brought to the main paper.
3. I think an analysis to show when the model works and more crucially when it doesn't work would be very important. The maximum utility would come in predicting areas which look like normal areas but have childhood poverty. Does the model do well in such areas? Similarly, one can analyse for the temporal analysis presented in the paper.

**Relation To Prior Work:**

There is a plethora of related work. I feel the paper does a decent job at relating to prior work. However, perhaps a table comparing the previous similar studies might make it easier for a reader to understand the scale, variables, etc. The authors do discuss it in their appendix. I have a slight feeling that it may be better to have more of that information in the main paper.

**Summary And Contributions:**

The paper contributes a dataset containing 33000+ satellite images of 10kmX10km resolution and corresponding labels across multiple categories corresponding to the child poverty variable. The dataset is a meaningful contribution owing to the: i) social problem at hand; ii) the size of the dataset; iii) diversity of labels and input data (19 countries).

---

> ### Author Rebuttal · Authors · 2024-08-16
>
> We thank the reviewer for their attention to our paper.
>
> You note: “the dataset part gets squeezed a bit”. We agree. Regarding the dataset size and diversity, we have added a sentence to section 3.2, "Demographic Health Surveys and Child Poverty":
>
> Our dataset consists of survey data for 275,921 children in 24,214 cluster locations across 16 countries in Eastern and Southern Africa for 1997-2022, with satellite images for each cluster location. Eastern and Southern Africa is a geographic region covering 13 million square km (40% of the African continent), with hundreds of languages, and significant diversity in socioeconomic and environmental conditions. Each survey dataset is nationally representative for a given year.
>
> (Note that the abstract incorrectly stated 19 countries; there are actually 20 countries in Eastern and Southern Africa, with DHS data available for 16 of them.)
>
> Regarding the dataset attributes, we have added a new table to the main paper that describes the indicators that make up child poverty and lists exactly which variables were used from the DHS survey. The table is attached.
>
> Finally, we agree that showing “when [the model] doesn't work would be very important“. As discussed in the response to the third reviewer, we plan to add a leave-one-country out analysis. We will discuss this analysis, and also follow your suggestion of adding text discussing the temporal analysis to section 5.3 “Interpretation of results”.

---

> > ### Comment · Reviewer_yepH · 2024-08-30
> > **Thank you for your rebuttal**
> >
> > Dear authors
> > I thank you for your comments to my questions. My score was already good and there is no fractional scores, else I might have increased scores to 7.5. I am looking for your responses to other comments.
> >
> > For now, all my concerns seem to be addressed; but I note other reviewers' comments.

---

### Official Review · Reviewer_inpU · 2024-07-23

**Rating:** 6
**Confidence:** 4
**Correctness:** Yes
**Clarity:** Yes

**Review:**

The research topic of using satellite data to estimate child poverty is very meaningful for social good and sustainable development.  The proposed dataset can be used to benchmark existing foundation models.

**Strengths:**

The dataset is well documented and the codebase is also provided with detailed instructions on how to use it.
The dataset is quite useful for sustainable development and could also be used in other social good-related research.
The authors provide benchmarking results of different models including foundation models and non-deep learning methods (GPR).

**Additional Feedback:**

N/A

**Documentation:**

Yes

**Opportunities For Improvement:**

My only concerns are about the benchmarking experiments part:

1. It would be nice if the authors could provide results of more deep learning models. Regarding foundation models, there are also lots of foundation models for remote sensing data missing.
2. Visualization results should be provided to give an intuitive understanding of the performance of different deep-learning models.

**Relation To Prior Work:**

Yes

**Summary And Contributions:**

The paper introduces a new dataset that pairs satellite imagery with high-quality survey data on child poverty to benchmark satellite feature representations. This dataset comprises 33,608 images from 19 countries in Eastern and Southern Africa, spanning the period from 1997 to 2022. The dataset aligns with UNICEF’s multidimensional child poverty definition, calculated from Demographic and Health Surveys (DHS) data. The authors test spatial and temporal generalization by evaluating on unseen locations and data beyond the training period. They benchmark multiple models, including MOSAIKS to deep learning foundation models, including DINOv2 and SatMAE. The paper also provides open-source code for building the satellite dataset, obtaining ground truth data from DHS, and running the various models for benchmarking.

---

> ### Author Rebuttal · Authors · 2024-08-16
>
> We thank the reviewer for their summary of the strengths of the paper, and for highlighting the importance of our work for social good and sustainable development.
>
> *Regarding the inclusion of more deep learning models*: we agree that there is room to be more comprehensive and exhaustive. But given trends in computer vision and remote sensing, we were most focused on the goal you have underlined: "The proposed dataset can be used to benchmark existing foundation models."
>
> To that end, we were not trying to develop a new state-of-the-art approach; rather, as a submission to the Datasets and Benchmarks Track, we wanted to establish a realistic baseline, collect a globally important dataset, and provide a benchmark for systematically evaluating new computer vision approaches to remote sensing.
>
> In order to fully realize this goal, and promote replicable and reproducible research, we have added a well-documented workflow to our GitHub repo with a loader for the KidSat dataset and model evaluation functionality to enable researchers to rigorously evaluate new deep learning methods, extend the dataset to new countries or indicators, or both. We demonstrate this workflow on an important baseline that we previously overlooked, ResNet50, achieving 0.2399 for the spatial benchmark.
>
> *Regarding a visualization*: we totally agree that we overlooked the importance of providing a graphical depiction of our work. We have included a new figure in the paper (attached as a PDF here) to compare Gaussian process regression using Kenya DHS 2022 data only, DINOv2 fine-tuned for the spatial task with predictions for Kenya in 2022, and DINOv2 fine-tuned for the temporal task  with predictions for Kenya in 2022. And as we discuss in our response to the third reviewer, we plan to include more visualizations in the supplementary materials.

---

### Decision · Program_Chairs · 2024-09-26

**Decision:**

Reject

**Comment:**

The AC thinks the dataset topic is very important and interesting, and quite novel with respect to the usual topics represented at the conference. This is very positive!
However, serval issues were raised about the experiments, the potential usages of the very rich data and the overall level of polishing. This AC believes many of these comments (in particular those raised by rwhX) make sense and addressing them explicitly would make a future submission much stronger. The AC has no doubt the paper will be published at the next round.